# GUIDED SAFE SHOOTING: MODEL BASED REINFORCEMENT LEARNING WITH SAFETY CONSTRAINTS

## ABSTRACT

In the last decade, reinforcement learning successfully solved complex control tasks and decision-making problems, like the Go board game. Yet, there are few success stories when it comes to deploying those algorithms to real-world scenarios. One of the reasons is the lack of guarantees when dealing with and avoiding unsafe states, a fundamental requirement in critical control engineering systems. In this paper, we introduce Guided Safe Shooting (GuSS), a model-based RL approach that can learn to control systems with minimal violations of the safety constraints. The model is learned on the data collected during the operation of the system in an iterated batch fashion, and is then used to plan for the best action to perform at each time step. We propose three different safe planners, one based on a simple random shooting strategy and two based on MAP-Elites, a more advanced divergent-search algorithm. Experiments show that these planners help the learning agent avoid unsafe situations while maximally exploring the state space, a necessary aspect when learning an accurate model of the system. Furthermore, compared to model-free approaches, learning a model allows GuSS reducing the number of interactions with the real-system while still reaching high rewards, a fundamental requirement when handling engineering systems.

## 1 INTRODUCTION

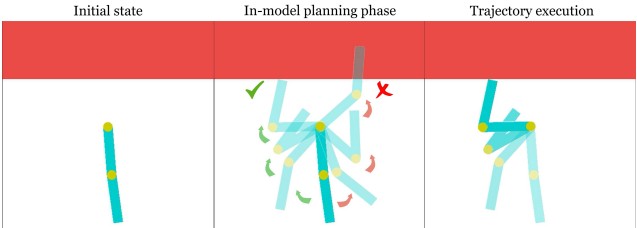

Figure 1: An illustrative example of planning with model-based approach on the Acrobot environment. The agent controls the torque on the first joint with the goal of getting its end effector as high as possible, avoiding the unsafe zone (red area). Starting in the rest position (left) the agent uses its model to find the best plan (middle) that will maximize the reward while satisfying the safety constraint and execute it on the real system (right). The example is especially relevant to applications in which safety and reward are traded off.

In recent years, deep Reinforcement Learning (RL) solved complex sequential decision-making problems in a variety of domains, such as controlling robots, and video and board games (Mnih et al., 2015; Andrychowicz et al., 2020; Silver et al., 2016). However, in the majority of these cases, success is limited to a simulated world. The application of these RL solutions to real-world systems is still yet to come. The main reason for this gap is the fundamental principle of RL of learning by trial and error to maximize a reward signal (Sutton & Barto, 2018). This framework requires unlimited access to the system to explore and perform actions possibly leading to undesired outcomes. This is not always possible. For example, considering the task of finding the optimal control strategy for a data center cooling problem (Lazic et al., 2018), the RL algorithm could easily take actions leading to high temperatures during the learning process, affecting and potentially breaking the

system. Another domain where safety is crucial is robotics. Here unsafe actions could not only break the robot but could potentially also harm humans. This issue, known as *safe exploration*, is a central problem in AI safety (Amodei et al., 2016). This is why most achievements in RL are in simulated environments, where the agents can explore different behaviors without the risk of damaging the real system. However, those simulators are not always accurate enough, if available at all, leading to suboptimal control strategies when deployed on the real-system (Salvato et al., 2021).

With the long-term goal of deploying RL algorithms on real engineering systems, it is imperative to overcome those limitations. A straightforward way to address this issue is to develop algorithms that can be deployed directly on the real system that provide guarantees in terms of constraints, such as safety, to ensure the integrity of the system. This could potentially have a great impact, as many industrial systems require complex decision-making, which efficient RL systems can easily provide. Going towards this goal, in this paper we introduce Guided Safe Shooting (GuSS), a safe Model Based Reinforcement Learning (MBRL) algorithm that learns a model of the system and uses it to plan for a safe course of actions through Model Predictive Control (MPC) (Garcia et al., 1989). GuSS learns the model in an iterated batch fashion (Matsushima et al., 2021; Kégl et al., 2021), allowing for minimal real-system interactions. This is a desirable property for safe RL approaches, as fewer interactions with the real-system mean less chance of entering unsafe states, a condition difficult to attain with model-free safe RL methods (Achiam et al., 2017; Ray et al., 2019b; Tessler et al., 2018). Moreover, by learning a model of the system, this allows flexibility and safety guarantees as using the model we can anticipate unsafe actions before they occur. Consider the illustrative example in Fig. 1: the agent, thanks to the model of its dynamics, can perform "mental simulation" and select the best plan to attain its goal while avoiding unsafe zones. This contrasts with many of the methods in the literature that address the problem of finding a safe course of action through Lagrangian optimization or by penalizing the reward function (Webster & Flach, 2021; Ma et al., 2021; Cowen-Rivers et al., 2022). GuSS avoids unsafe situations by discarding trajectories that are deemed unsafe using the model predictions. Within this framework, we propose three different safe planners, one based on a simple random shooting strategy and two based on MAP-Elites (ME) (Mouret & Clune, 2015), a more advanced Quality-Diversity (QD) algorithm. These planners are used to generate, evaluate, and select the safest actions with the highest rewards. Using divergent-search methods for planning allows the agent to more widely explore the possible courses of actions. This leads to both a safer and more efficient search, while covering a higher portion of the state space, an important factor when learning a model, given that more exploratory data lead to better models (Yarats et al., 2022).

We test GuSS on three different environments. The presented results highlight how the model and planners can easily find strategies reaching high rewards with minimal costs, even when the two metrics are antithetical, as is the case for the Safe Acrobot environment.

To recap, the contributions of the paper are the following:

- We introduce Guided Safe Shooting (GuSS), an MBRL method capable of efficiently learning to avoid unsafe states while optimizing the reward;
- We propose the use of quality-diversity evolutionary methods as MAP-Elites (ME) as planning techniques in MBRL approaches;
- We present 3 different planners, Safe Random Shooting (S-RS), Safe MAP-Elites (S-ME), Pareto Safe MAP-Elites (PS-ME), that can generate a wide array of action sequences while discarding the ones deemed unsafe during planning.

## 2 RELATED WORK

Some of the most common techniques addressing safety in RL rely on solving a Constrained Markov Decision Process (CMDP) (Altman, 1999) through model-free RL methods (Achiam et al., 2017; Ray et al., 2019b; Tessler et al., 2018; Hsu et al., 2021; Zhang et al., 2020). Among these approaches, a well-known method is CPO (Achiam et al., 2017) which adds constraints to the policy optimization process in a fashion similar to TRPO (Schulman et al., 2015). A similar approach is taken by PCPO (Yang et al., 2020b) and its extension (Yang et al., 2020a). The algorithm works by first optimizing the policy with respect to the reward and then projecting it back on the constraint set in an iterated two-step process. A different strategy consists in storing all the "recovery" actions that the agent took to leave unsafe regions in a separate replay buffer (Hsu et al., 2021). This buffer is used whenever

the agent enters an unsafe state by selecting the most similar transition in the safe replay buffer and performing the same action to escape the unsafe state.

Model-free RL methods need many interactions with the real-system in order to collect the data necessary for training. This can be a huge limitation in situations in which safety is critical, where increasing the number of samples increases the probability of entering unsafe states. MBRL limits this problem by learning a model of the system that can then be used to learn a safe policy. This allows increased flexibility in dealing with unsafe situations, even more if the safety constraints change in time.

Many of these methods work by modifying either the cost or the reward function to push the algorithm away from unsafe areas. The authors of Uncertainty Guided Cross-Entropy Methods (CEM) (Webster & Flach, 2021) extend PETS (Chua et al., 2018) by modifying the objective function of the CEM-based planner to avoid unsafe areas. In this setting, an unsafe area is defined as the set of states for which the ensemble of models has the highest uncertainty. A different strategy is to inflate the cost function with an uncertainty-aware penalty function, as done in CAP (Ma et al., 2021). This cost change can be applied to any MBRL algorithm and its conservativeness is automatically tuned through the use of a PI controller. Another approach, SAMBA (Cowen-Rivers et al., 2022), uses Gaussian Processes (GP) to model the environment. This model is then used to train a policy by including the safety constraint in the optimization process through Lagrangian multipliers. Closer to GuSS are other methods using a trajectory sampling approach to select the safest trajectories generated by CEM (Wen & Topcu, 2018; Liu et al., 2020). This lets us deal with the possible uncertainties of the model predictions that could lead to consider a trajectory safe when it is not.

## 3 BACKGROUND

In this section, we introduce the concepts of safe-RL and QD algorithms on which our method builds.

### 3.1 SAFE REINFORCEMENT LEARNING

Reinforcement learning problems are usually represented as a Markov decision process (MDP) $\mathcal{M} = \langle \mathcal{S}, \mathcal{A}, \mathcal{T}, r, \gamma \rangle$, where $\mathcal{S}$ is the state space, $\mathcal{A}$ is the action space, $\mathcal{T} : \mathcal{S} \times \mathcal{A} \to \mathcal{S}$ is the transition dynamics, $r : \mathcal{S} \times \mathcal{A} \to \mathbb{R}$ is the reward function and $\gamma \in [0, 1]$ is the discount factor. Let $\Delta(\mathcal{X})$ denote the family of distributions over a set $\mathcal{X}$. The goal is to find a policy $\pi : \mathcal{S} \to \Delta(\mathcal{A})$ which maximizes the expected discounted return $\pi^* = \arg\max_{\pi} \mathbb{E}_\pi \left[ \sum_t \gamma^t r(s_t, a_t) \right]$ (Sutton & Barto, 2018). This formulation can be easily accommodated to incorporate constraints, for example representing safety requirements. To do so, similarly to the reward function, we define a new cost function $\mathcal{C} : \mathcal{S} \times \mathcal{A} \to \mathbb{R}$ which, in our case, is a simple indicator for whether an unsafe interaction has occurred ($\mathcal{C}_t = 1$ if the state is unsafe and $\mathcal{C}_t = 0$ otherwise). The new goal is then to find an optimal policy $\pi^*$ with a high expected reward $\mathbb{E}_\pi \left[ \sum_t \gamma^t r(s_t, a_t) \right]$ and a low safety cost $\mathbb{E}_\pi \left[ \sum_t C(s_t, a_t) \right]$. One way to solve this new problem is to rely on constrained Markov Decision processes (CMDPs) (Altman, 1999) by adding constraints on the expectation (La & Ghavamzadeh, 2013) or on the variance of the return (Chow et al., 2017).

### 3.2 MODEL BASED REINFORCEMENT LEARNING

In this work, we address the issue of respecting safety constraints through an MBRL approach (Moerland et al., 2021). In this setting, the transition dynamics $p_{\text{real}}$ are estimated using the data collected when interacting with the real system. The objective is to learn a model $p(s_t, a_t) \rightsquigarrow s_{t+1}$[1] to predict $s_{t+1}$ given $s_t$ and $a_t$ and use it to learn an optimal policy $\pi^*$. In this work, we considered the iterated-batch learning approach (also known as growing batch (Lange et al., 2012) or semi-batch (Singh et al., 1994)). In this setting, the model is trained and evaluated on the real-system through an alternating two-step process consisting in: (i) applying and evaluating the learned policy on the environment for a whole episode and (ii) then training the model on the growing data of transitions collected during the evaluation $\tau$ itself $\mathcal{T}r_t = \{(s_1, a_1, r_1, c_1, s'_1)^0, \ldots, (s_t, a_t, r_t, c_t, s'_t)^0, \ldots, (s_1, a_1, r_1, c_1, s'_1)^\tau, \ldots, (s_t, a_t, r_t, c_t, s'_t)^\tau\}$

---

[1]We use $\rightsquigarrow$ to denote both probabilistic and deterministic mapping.

and updating the policy. The process is repeated until a given number of evaluations or a certain model precision is reached.

## 3.3 QUALITY DIVERSITY

QD methods are a family of Evolution Algorithms (EAs) performing divergent search with the goal of generating a collection of diverse but highly performing policies (Pugh et al., 2016; Cully & Demiris, 2017). The divergent search is performed over a, usually hand-designed, *behavior space* $\mathcal{B}$ in which the behavior of the evaluated policies is represented. Some of these methods have been shown, given enough iterations, to be able to uniformly cover the whole behavior space (Doncieux et al., 2019). Combining these methods with RL approaches then allows to greatly increase the exploration abilities of RL algorithms.

Among the different QD methods introduced in the literature (Lehman & Stanley, 2011; Paolo et al., 2021; Mouret & Clune, 2015), in this work we use the ME algorithm (Mouret & Clune, 2015) due to its simplicity and power. A detailed description of how ME works and the related pseudo-code are presented in Appendix A.

## 4 METHOD

In this work, we use a model of the environment to plan safe trajectories maximizing the reward function. In many real settings it is possible to assume that the set of unsafe situations is specified by the engineer. For this reason we consider that the cost function is given and the cost is calculated from the states the system is in.

In this section, we describe in detail how GuSS trains the model and uses the planners. An overview of the method is presented in Alg. 1. The code is available at: `<URL hidden for review>`.

---

**Algorithm 1:** Guided Safe Shooting (GuSS)

---
1  **INPUT:** real-system $p_{\text{real}}$, number of episodes $M$, number of action sequences for planning step $N$, planning horizon length $h$, episode length $T$, initial random policy $\pi^0$;
2  **RESULT:** learned model $p(\cdot)$ and planner;
3  Initialize empty trace $\mathcal{T}r = \varnothing$;
4  $s_0 \hookleftarrow p_{\text{real}}$                      ▷ Sample initial state from real system
5  **for** *t in [0, ..., T]* **do**
6       $a_t \hookleftarrow \pi^0(s_t)$                 ▷ Generate random action
7       $\mathcal{T}r = \mathcal{T}r \bigcup (a_t, s_t)$            ▷ Store transition in trace
8       $s_{t+1} \hookleftarrow p_{\text{real}}(s_t, a_t)$        ▷ Apply action on real system and get new state
9  **for** *i in [1,...,M]* **do**
10      $p^i \leftarrow \text{TRAIN}(p^{i-1}, \mathcal{T}r)$           ▷ Train model
11      **for** *t in [0, ..., T]* **do**
12          $a_t \hookleftarrow \text{PLAN}(p^i, s_t, h, N)$      ▷ Use planner to generate next action
13          $\mathcal{T}r = \mathcal{T}r \bigcup (a_t, s_t)$        ▷ Store transition in trace
14          $s_{t+1} \hookleftarrow p_{\text{real}}(s_t, a_t)$     ▷ Apply action on real system and get new state

---

## 4.1 TRAINING THE MODEL

Let $\mathcal{T}r_t = \{(s_1, a_1, r_1, c_1, s_1')...(s_t, a_t, r_t, c_t, s_t')\}$ be a system trace consisting of $t$ steps and $(s_t, a_t)$ a state-action tuple. The goal is to train a model $p$ to predict $s_{t+1}$ given the previous $(s_t, a_t)$. This model can be learned in a supervised fashion given the history trace $\mathcal{T}r_t$. We chose as $p$ a deterministic deep mixture density network (Bishop, 1994), which has proven to have good properties when used in MPC (Kégl et al., 2021). More details on the model parameters and training can be found in Appendix E.

## 4.2 Planning for safety

The trained model $p(s_t, a_t)$ is used in an MPC fashion to select the action $a_t$ to apply on the real system. This means that at every time step $t$ we use the model to evaluate $N$ action sequences by simulating the trajectories of length $h$ from state $s_t$. For each action sequence, the return $R$ and the cost $C$

$$R = \sum_{k=0}^{h} \gamma^k r(\tilde{s_k}, a_k), \quad C = \sum_{k=0}^{h} \gamma^k c(\tilde{s_k}, a_k); \quad (1)$$

are evaluated, where $\tilde{s_k}$ is the state generated by the model $p(\cdot)$ and $\gamma$ the discount factor. GuSS then selects the first action of the best sequence, with respect to both the return and the cost. In this work, we assume that both the reward function $r(s, a)$ and the cost function $c(s, a)$ are given. In reality, this is not limiting, as in many real engineering settings both the reward and the cost are known and given by the engineer. We tested three different approaches to generate and evaluate safe action sequences at planning time.

### 4.2.1 Safe Random Shooting (S-RS)

Based on the Random Shooting planner used in Kégl et al. (2021), Safe Random Shooting (S-RS) generates $N$ random sequences of actions $a_i \in A$ of length $h$. These sequences are then evaluated on the model starting from state $s_t$. The next action $a_t$ to apply on the real system is selected from the action sequence with the lowest cost. If multiple action sequences have the same cost, $a_t$ is selected from the one with the highest reward among them. The pseudocode of the planner is shown in Appendix B.1.

### 4.2.2 Safe MAP-Elites (S-ME)

Safe MAP-Elites (S-ME) is a safe version of the ME algorithm detailed in Appendix A. Rather than directly generating sequences of actions as done by S-RS, here we generate the weights $\phi$ of small Neural Networks (NNs) that are then used to generate actions depending on the state provided as input: $\phi(s_t) = a_t$. This removes the dependency on the horizon length $h$ of the size of the search space present in S-RS. After the evaluation of a policy $\phi_i$, this is added to the collection $\mathcal{A}_{ME}$. If another policy with the same behavior descriptor has already been found, S-ME only keeps the policy with the lowest cost. If the costs are the same, the one with the highest reward will be stored.

Moreover, at each generation, the algorithm samples $K$ policies $\phi$ from the collection to generate $K$ new policies $\tilde{\phi}$. For this step, only the policies with $C = 0$ are considered. If there are enough policies in the collection satisfying this requirement, the probability of sampling each policy is weighted by its reward. On the contrary, if only $k < K$ policies with $C = 0$ are in the collection, the missing $K - k$ are randomly generated. This increases the exploration and can be useful in situations in which it is difficult not to incur in any cost. The pseudocode of the planner is shown in Appendix B.2.

### 4.2.3 Pareto Safe MAP-Elites (PS-ME)

Another safe version of the ME algorithm. Contrary to S-ME, which only samples policies for which $C = 0$, PS-ME sorts all the policies present in the collection into non-dominated fronts. The $K$ policies are then sampled from the best non-dominated front. In case less than $K$ policies are present on this front, PS-ME samples them from the other non-dominated front, in decreasing order of non-domination, until all $K$ policies are selected.

This strategy takes advantage of the search process operated until that point even when not enough safe solutions are present, rather than relying on random policies as done with S-ME. The pseudo code of the planner is shown in Appendix B.3.

## 5 Experiments

### 5.1 Environments

We test GuSS on two different OpenAI gym environments with safety constraints (pendulum swing-up, Acrobot) and the OpenAI SafeCar-Goal safety-gym environment (Ray et al., 2019a). In the

environment design we follow previous work (Cowen-Rivers et al., 2022; Ray et al., 2019b) and delegate the details to Appendix C. Moreover, we use the stochastic version of the SafeCar-Goal environment with the position of the unsafe areas randomly resampled at the beginning of each episode. This makes the environment much harder, not allowing the agents to overfit the position of the unsafe zones. To our knowledge, we are the first to use this version to test an MBRL approach, where other works focused on the easier version with no layout randomization used by Yang et al. (2021).

## 5.2 RESULTS

We compare GuSS with the three different planners introduced in Sec. 4.2 against various baselines. To have a baseline about how much different are the performances of safe methods with respect to unsafe ones, we compared against two unsafe versions of GuSS: **RS** and **ME**. **RS** performs random shooting to plan for the next action, while **ME** uses vanilla MAP-Elites as a planner, without taking into account any safety requirement. We also compared against the safe MBRL approach **RCEM** (Liu et al., 2020) and its respective unsafe version, labeled **CEM**. Moreover, to show the efficiency of model-based approaches when dealing with safety requirements, we compared against three model-free baselines: **CPO** (Achiam et al., 2017), **TRPO lag** and **PPO lag**; all of them come from the Safety-Gym benchmark (Ray et al., 2019b).

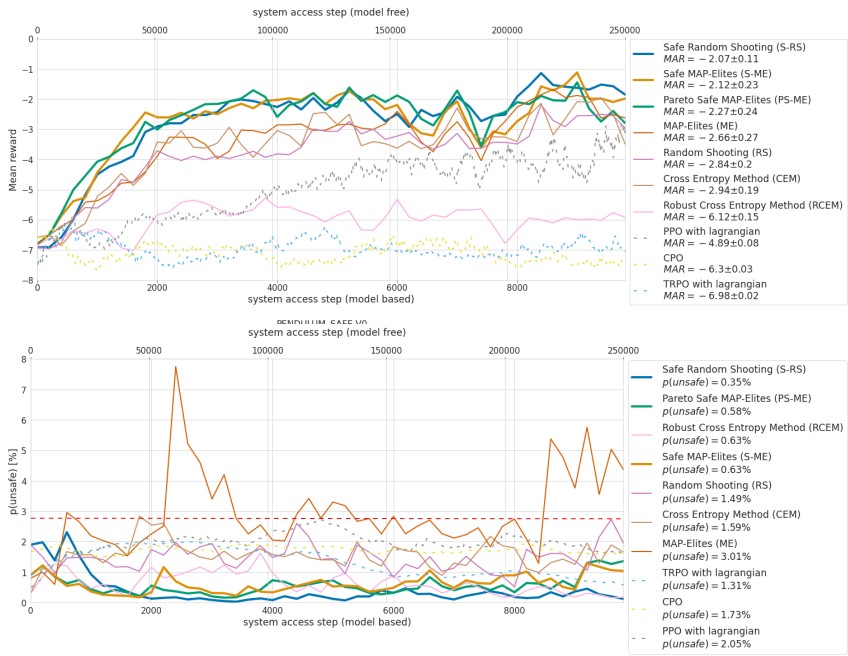

Figure 2: Mean reward and probability percentage of unsafe for Safe Pendulum environment. Dashed curves indicate Model-free baselines and plain one Model-based approaches. Thicker lines represent the proposed algorithms. The red dashed line indicates the random unsafe probability. All curves are calculated over 5 random seed.

The algorithms are compared according to four metrics: Mean Asymptotic Reward (**MAR**), Mean Reward Convergence Pace (**MRCP**), Probability percentage of unsafe (**p(unsafe)[%]**) and transient probability percentage of unsafe (**p(unsafe)[%]**$_{trans}$). The details on how these metrics are calculated are defined in Appendix D.

The results are shown in Table 1. The MAR scores and the $p(\text{unsafe})[\%]$ for the pendulum system are shown in Fig. 2, while the ones for the acrobot system are in Fig. 3. Finally, Fig. 4 shows the results for the SafeCar-Goal environment. Additional plots are presented in Appendices F and G.

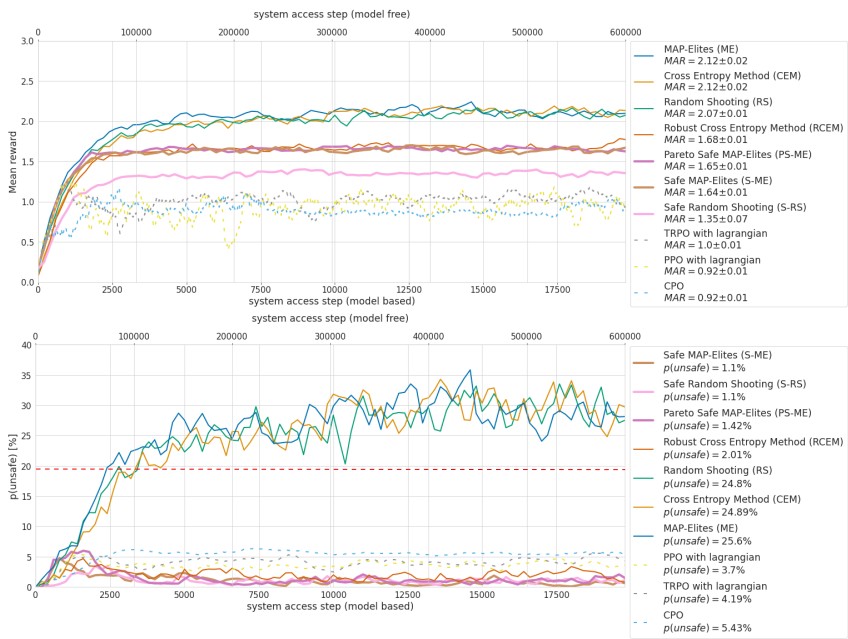

Figure 3: Mean reward and probability percentage of unsafe for Safe Acrobot environment. Dashed curves indicate Model-free baselines and plain one Model-based approaches. Thicker lines represent the proposed algorithms. The red dashed line indicates the random unsafe probability. All curves are calculated over 5 random seed.

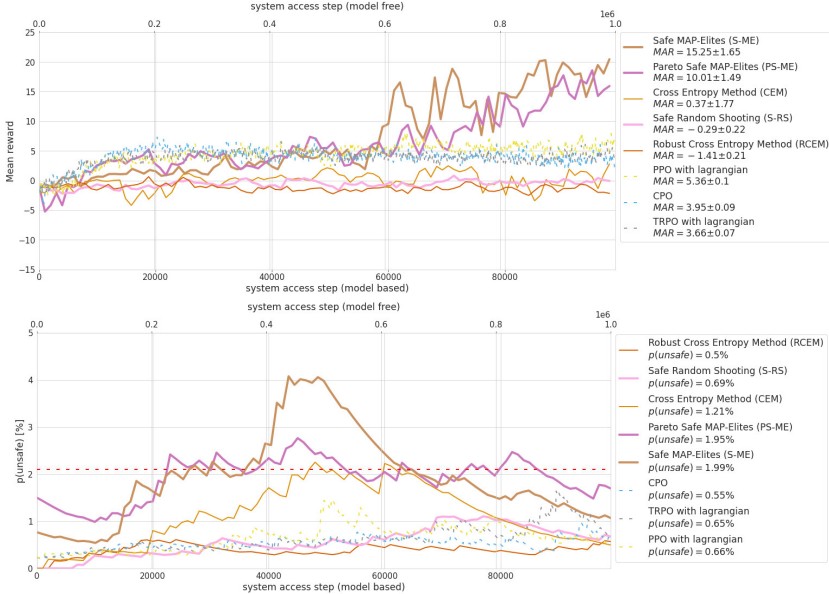

Figure 4: Mean reward and probability percentage of unsafe for SafeCar-Goal environment. Dashed curves indicate Model-free baselines and plain one Model-based approaches. Thicker lines represent the proposed algorithms. The red dashed line indicates the random unsafe probability. All curves are calculated over 3 random seed.

# 6  DISCUSSION

The results presented in Sec. 5.2 show how GuSS can reach high performances while keeping safety cost low on two of the three environments tested. As expected, on Acrobot, safe methods reach

Table 1: Summary of the different methods on the two different environments Pendulum and Acrobot. MAR is the mean asymptotic reward, MRCP($r_{thr}$) is the number of system access steps needed to achieve $r_{thr}$ of the optimum reward, $p$(unsafe)[%] and $p$(unsafe)[%]$_{trans}$ are the probability percentage of being unsafe during epochs. All the metrics are average over all epochs and seeds and ↓ and ↑ mean lower and higher the better, respectively. The best **safe methods** with respect to each metric are highlighted in bold. All ± values are 90% Gaussian confidence interval.

| Method | MAR ↑ | MRCP×$10^3$ ↓ | $p$(unsafe)[%] ↓ | $p$(unsafe)$_{trans}$[%] ↓ |
|---|---|---|---|---|
| | Safe Pendulum $r_{thr} = -2.5$ | | | |
| GuSS(S-RS) | **-2.09** ± **0.09** | 2.12 ± 1.45 | **0.35** ± **0.63** | 1.23 ± 1.28 |
| GuSS(S-ME) | -2.2 ± 0.16 | **2.0** ± **0.9** | 0.63 ± 0.57 | **0.54** ± **0.57** |
| GuSS(PS-ME) | -2.27 ± 0.17 | 2.10 ± 1 | 0.58 ± 0.52 | 0.6 ± 0.56 |
| RCEM | -6.12 ± 0.15 | 6.7 ± 0.81 | 0.63 ± 0.64 | 0.84 ± 0.7 |
| RS | -2.7 ± 0.14 | 2.60 ± 1.2 | 1.49 ± 1.0 | 1.36 ± 0.61 |
| ME | -2.53 ± 0.19 | 1.90 ± 0.7 | 3.01 ± 2.78 | 2.01 ± 1.95 |
| CEM | -2.99 ± 0.15 | 1.64 ± 0.41 | 1.59 ± 0.89 | 1.43 ± 0.85 |
| CPO | -6.06 ± 0.04 | 22 ± 0.0 | 1.73 ± 0.92 | 1.59 ± 0.78 |
| PPO lag | -4.1 ± 0.12 | 138 ± 65 | 2.05 ± 1.75 | 2.15 ± 1.48 |
| TRPO lag | -7.02 ± 0.03 | 161 ± 55 | 1.31 ± 0.91 | 2.07 ± 0.89 |
| | Safe Acrobot $r_{thr} = 1.6$ | | | |
| GuSS(S-RS) | 1.35 ± 0.07 | 1.6 ± 0.26 | **1.1** ± **1.05** | **1.85** ± **1.56** |
| GuSS(S-ME) | 1.64 ± 0.01 | **1.36** ± **0.25** | **1.1** ± **1.22** | 2.45 ± 1.92 |
| GuSS(PS-ME) | 1.65 ± 0.01 | 1.56 ± 0.11 | 1.42 ± 1.62 | 3.75 ± 2.77 |
| RCEM | **1.68** ± **0.01** | 1.60 ± 0.37 | 2.01 ± 1.4 | 2.85 ± 1.84 |
| RS | 2.07 ± 0.01 | 1.28 ± 0.29 | 24.8 ± 8.3 | 10.85 ± 7.36 |
| ME | 2.12 ± 0.02 | 1.12 ± 0.24 | 25.6 ± 7.91 | 12.18 ± 7.95 |
| CEM | 2.12 ± 0.02 | 1.40 ± 0.43 | 24.9 ± 8.91 | 9.14 ± 7.17 |
| CPO | 0.94 ± 0.01 | 87 ± 59 | 5.43 ± 1.74 | 4.35 ± 2.49 |
| PPO lag | 0.94 ± 0.01 | 24 ± 3 | 3.7 ± 1.94 | 4.02 ± 2.46 |
| TRPO lag | 1.02 ± 0.01 | 37 ± 22 | 4.2 ± 2.0 | 4.31 ± 2.53 |
| | SafeCar-Goal $r_{thr} = 10$ | | | |
| GuSS(S-RS) | -0.29 ± 0.22 | - ± - | 0.69 ± 1.04 | 0.49 ± 0.83 |
| GuSS(S-ME) | **15.25** ± **1.65** | 32 ± 0.0 | 1.99 ± 3.52 | 0.55 ± 0.9 |
| GuSS(PS-ME) | 10.01 ± 1.49 | **12.33** ± **4.54** | 1.95 ± 2.71 | 0.76 ± 1.10 |
| RCEM | -1.41 ± 0.21 | - ± - | **0.50** ± **0.83** | 0.63 ± 0.99 |
| CEM | 0.37 ± 1.77 | 58 ± 9.31 | 1.21 ± 2.47 | 0.67 ± 1.26 |
| CPO | 3.95 ± 0.09 | 297 ± 153.6 | 0.55 ± 1.11 | 0.42 ± 0.68 |
| PPO lag | 5.36 ± 0.1 | 231 ± 121.24 | 0.66 ± 1.51 | **0.34** ± **0.58** |
| TRPO lag | 3.66 ± 0.07 | 108 ± 0.0 | 0.65 ± 1.58 | 0.39 ± 0.72 |

lower MAR scores compared to unsafe ones due to these last ones ignoring the safety constraints. At the same time, this leads to much higher $p$(unsafe) for unsafe approaches. Interestingly, this is not the case on Pendulum where unsafe approaches tend to have lower MAR scores compared to safe methods even while not respecting safety criteria. The best unsafe method, ME, has in fact a MAR of $-2.53 \pm 0.19$ while the worst MBRL safe method, PS-ME, has a MAR of $-2.27 \pm 0.17$ (p $< 1e - 08$). This is likely due to the unsafe region effectively halving the search space. The left hand side in fact provides lower "safe rewards" compared to the right hand side, pushing the safe algorithms to focus more on swinging the arm towards the right than the left. This is particularly visible in Appendices F between the safe and unsafe methods where the unsafe area cleanly cuts the distribution of visited states. Moreover, while the safe methods, and in particular ME cover a larger part of the state space, the safe ones tend to focus much more on the high rewarding state.

When comparing the performances of the different safe planners for GuSS, it is possible to notice how the simpler S-RS planner outperforms the more complex S-ME and PS-ME on Pendulum with respect to both MAR ($p < 0.02$) and $p(\text{unsafe})$ ($p < 0.05$). This is not the case on the more complex Acrobot, where the improved search strategies performed by S-ME and PS-ME reach higher MAR scores compared to S-RS ($p < 0.02$), while being comparable with respect to the safety cost. This hints at the fact that while a simple strategy like S-RS is very efficient in simple set-ups, more advanced planning techniques are needed in more complex settings. RCEM the closest method to GuSS, failed on Pendulum but reached higher MAR on Acrobot, however with twice the safety cost.

The need for more complex planners is even more evident on the hardest environment we tested: SafeCar-Goal. Here, S-ME and PS-ME clearly outperforms all other methods, even the model-free approaches, in terms of MAR. At the same time, both methods show to be the most unsafe. In fact, while on the $p(\text{unsafe})_{trans}$ metric S-ME and PS-ME perform similarly to the model-free approaches, this is not the case for the $p(\text{unsafe})$ which increases to surpass the random unsafe probability before decreasing to acceptable levels. This can be explained by the increased complexity of the environment due to the layout randomization of the unsafe areas at the beginning of each episode, which requires for the model to generalize in order to safely navigate the environment. This effect is supported looking at S-ME and PS-ME where respectively at around 50k steps $p(\text{unsafe})$ drops while the reward increases. A similar effect can be observed with CEM but here there is no reward increase. Moreover, while the high $p(\text{unsafe})$ during the training seems to be consistent with simply ignoring the safety, closer inspection shows that this is not the case. The planning agent mostly avoids unsafe areas, with the exception of when it does an error and traverses them. In that case, it tends to get stuck on them without being able to leave. This is possible due to the nature of the observables inside the unsafe regions and requires a more in-depth investigation.

At the same time, while model-free methods tend to remain safer than GuSS, they require at least an order of magnitude more real-system interactions while still obtaining low MAR scores.

These results seem to confirm how the increased exploration of QD methods helps with the collection of more informative data, crucial when learning the model in the MBRL setting. This is particularly visible in the Safety-gym environment where no other baselines methods manage to solve the environment.

## 7 CONCLUSION AND FUTURE WORK

In this study, we proposed GuSS, a model based planning method for safe reinforcement learning. We tested the method on three environments with safety constraints and the results show that, while being simple, GuSS provides good results in terms of both safety and reward trade-off with minimal computational complexity. Moreover, we observed that in some settings, like Safe Pendulum, the introduction of safety constraints can lead to a better completion of the task. Further experiments are needed to confirm this effect, but if this holds, it could be possible to take advantage of it in a curriculum learning fashion: starting from strict safety constraints and incrementally relaxing them to get to the final task and safety requirement. This could help even more in the application of RL methods to real engineering systems, allowing the engineers handling the system to be more confident in the performances of the methods.

Notwithstanding the great results obtained by our proposed approach, the performances of GuSS are still tied to the accuracy of the model. If the model is wrong, it could easily lead the agent to unsafe states. This is particularly noticeable in the SafeCar-Goal environment, where the unsafe areas layout randomization renders the environment particularly hard for model based approaches. This led our method to obtain high reward but also high cost. As discussed in Section 6, these results require more in depth analysis, but also open the path to new directions of research. A possible solution to the problem of the model inaccuracy is to have uncertainty aware models that can predict their own uncertainty on the sampled trajectories and use it to be more conservative in terms of safety when learning the model. At the same time, the ME-based safe planners we proposed also suffer from the limitation of many QD approaches, namely the need to hand-design the behavior space. While some works have been proposed to address this issue (Cully, 2019; Paolo et al., 2020), these add another layer of complexity to the system, possibly reducing performance with respect to the safety constraints.

## Reproducibility Statement

In order to ensure reproducibility we will release the code at `<URL hidden for review>`, once the paper has been accepted. Moreover, the code is also attached to the submission as supplementary material.

Finally, all the hyperparameters of the algorithms are listed in Appendix E and the detailed pseudocode is shown in Appendix B.

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

# A  MAP-ELITES

In MAP-Elites (ME) the behavior space $\mathcal{B}$ is discretized through a grid with the algorithm trying to fill every cell of the grid (Mouret & Clune, 2015). ME starts by sampling the parameters $\phi \in \Phi$ of $M$ policies from a random distribution and evaluating them in the environment. The behavior descriptor $b_i \in \mathcal{B}$ of a policy $\phi_i$ is then calculated from the sequence of states traversed by the system during the policy evaluation. This descriptor is then assigned to the corresponding cell in the discretized behavior space. If no other policy with the same behavior descriptor is discovered, $\phi_i$ is stored as part of the collection of policies $\mathcal{A}_{\text{ME}}$ returned by the method. On the contrary, the algorithm only stores, in the collection, the policy with the highest reward among those with the same behavior descriptor. This allows the gradual increase of the quality of the policies stored in $\mathcal{A}_{\text{ME}}$. At this point, ME randomly samples a policy from the collection, and uses it to generate a new policy $\tilde{\phi}_i$ to evaluate. The generation of $\tilde{\phi}_i$ is done by adding random noise to its parameters through a *variation function* $\mathbb{V}(\cdot)$. The cycle repeats until the given evaluation budget $N$ is depleted. The pseudo-code of ME is shown in Alg. 2.

---

**Algorithm 2:** MAP-Elites

---

1   **INPUT:** real system $p_{\text{real}}$, initial state $s_0$, evaluation budget $N$, parameter space $\Phi$, discretized
      behavior space $\mathcal{B}$, variation function $\mathbb{V}(\cdot)$, number of initial policies $M$, episode length $H$;

2   **RESULT:** collection of policies $\mathcal{A}_{\text{ME}}$;

3   $\mathcal{A}_{\text{ME}} = \varnothing$                                      $\triangleright$ Initialize empty collection

4   $\Gamma \leftarrow \text{SAMPLE}(\Phi, M)$                           $\triangleright$ Sample initial $M$ policies

5   $R, C, \mathcal{T}_{real} = \text{ROLLOUT}(p_{\text{real}}, s_0, \phi_i, H), \quad \forall \phi_i \in \Gamma$      $\triangleright$ Evaluate policies in the real system

6   $b_i = f(\phi_i, \mathcal{T}^i_{real}), \quad \forall \phi_i \in \Gamma$ and $\forall \mathcal{T}^i_{real} \in \mathcal{T}_{real}$     $\triangleright$ Calculate policies behavior descriptors

7   $\mathcal{A}_{\text{ME}} \leftarrow \phi_i, \quad \forall \phi_i \in \Gamma$                       $\triangleright$ Store policies in collection

8   **while** $N$ *not depleted* **do**

9      |   $\Gamma \leftsquigarrow \text{SELECT}(\mathcal{A}_{\text{ME}}, K, \Phi)$             $\triangleright$ Select $K$ policies from collection

10    |   $\tilde{\Gamma} \leftsquigarrow \mathbb{V}(\Gamma)$                          $\triangleright$ Generate new policies

11    |   $R, C, \mathcal{T}_{real} = \text{ROLLOUT}(p_{\text{real}}, s_0, \tilde{\phi}_i, H), \quad \forall \tilde{\phi}_i \in \tilde{\Gamma}$    $\triangleright$ Evaluate new policies in the real
         |   system

12    |   $b_i = f(\tilde{\phi}_i, \mathcal{T}^i_{real}), \quad \forall \tilde{\phi}_i \in \tilde{\Gamma}$ and $\forall \mathcal{T}^i_{real} \in \mathcal{T}_{real}$    $\triangleright$ Calculate behavior descriptors

13    |   $\mathcal{A}_{\text{ME}} \leftarrow \tilde{\phi}_i, \quad \forall \tilde{\phi}_i \in \tilde{\Gamma}$                 $\triangleright$ Store policies in collection

---

# B PLANNERS PSEUDOCODE

This appendix contains the pseudo code of the three planners introduced in the paper. Each planner evaluates a policy on the model through a ROLLOUT function detailed in Alg. 3.

---
**Algorithm 3: ROLLOUT**

---
1 **INPUT:** model $p$, initial state $s_0$, policy $\phi$, horizon length $h$;
2 **RESULT:** collected reward $R$, collected cost $C$, simulated trace $\mathcal{T}_{model}$;
3 $\mathcal{T}_{model} = \varnothing$ $\qquad\qquad\qquad\qquad\qquad\qquad\qquad\qquad$ ▷ Initialize empty trace
4 **for** *j in [0, ..., h]* **do**
5 $\quad$ $a_j \twoheadleftarrow \phi(s_j)$ $\qquad\qquad\qquad\qquad\qquad\qquad$ ▷ Draw action from policy
6 $\quad$ $s_{j+1} \twoheadleftarrow p(s_j, a_j)$ $\qquad\qquad\qquad\qquad\qquad$ ▷ Draw next predicted state
7 $\quad$ $\mathcal{T}_{model} = \mathcal{T}_{model} \bigcup (s_j, a_j)$ $\qquad\qquad\qquad\qquad\qquad$ ▷ Update trace
8 $R = \sum_{j=0}^{h} \gamma^j r(s_j, a_j)$ $\qquad\qquad\qquad$ ▷ Calculate reward of action sequence
9 $C = \sum_{j=0}^{h} \gamma^j c(s_j, a_j)$ $\qquad\qquad\qquad$ ▷ Calculate cost of action sequence

---

## B.1 SAFE RANDOM SHOOTING

Algorithm 4 shows the pseudocode of the Safe Random Shooting (S-RS) planner.

---
**Algorithm 4: S-RS**

---
1 **INPUT:** model $p$, current real-system state $s_t$, planning horizon $h$, evaluated action sequences $N$, action space $\mathcal{A}$;
2 **RESULT:** action to perform $a_t$;
3 AS $= \varnothing$ $\qquad\qquad\qquad\qquad$ ▷ Initialize empty collection of action sequences
4 **for** *i in [0, ..., N]* **do**
5 $\quad$ $\phi_i = [a_t, \ldots, a_{t+h}] \twoheadleftarrow$ SAMPLE$(\mathcal{A}, h)$ $\qquad\qquad$ ▷ Sample action sequence
6 $\quad$ $R_i, C_i, \mathcal{T}_{model}^i =$ ROLLOUT$(p, s_t, \phi_i, h)$ $\qquad\qquad$ ▷ Evaluate action sequence
7 $\quad$ AS $=$ AS $\bigcup (\phi_i, R_i, C_i)$ $\qquad\qquad\qquad$ ▷ Store evaluated action sequence
8 AS$_{lc} \twoheadleftarrow$ AS $\qquad\qquad\qquad\qquad$ ▷ Get action sequences with lowest cost
9 $\phi_{best} \twoheadleftarrow$ AS$_{lc}$ $\qquad\qquad\qquad\qquad$ ▷ Get action sequence with highest reward
10 $a_t \twoheadleftarrow \phi_{best}(s_t)$ $\qquad\qquad\qquad\qquad\qquad\qquad$ ▷ Get next action

---

## B.2 SAFE MAP-ELITES

Algorithm 6 shows the pseudocode of the Safe MAP-Elites (S-ME) method. The SELECT function is shown in Alg. 5.

---
**Algorithm 5: SELECT function of S-ME planner**

---
1 **INPUT:** Collection of policies $\mathcal{A}_{ME}$, Number of policies to select $K$, Policy parameter space $\Phi$ ;
2 **RESULT:** set of selected policies $\Gamma$;
3 $\Gamma = \varnothing$ $\qquad\qquad\qquad\qquad\qquad\qquad$ ▷ Initialize empty set of selected policies
4 $\Gamma \twoheadleftarrow \mathcal{A}_{ME}[C = 0]$ $\qquad\qquad\qquad$ ▷ Select policies with $C = 0$ from collection
5 **if** *size*$(\Gamma) < K$ **then**
6 $\quad$ $\Gamma = \Gamma \bigcup$ SAMPLE$(\Phi)$ $\qquad\qquad$ ▷ Sample missing policies from parameter space

---

---

**Algorithm 6:** S-ME

1 **INPUT:** model $p$, current real-system state $s_t$, planning horizon $h$, evaluated action sequences $N$, discretized behavior space $\mathcal{B}$, variation function $\mathbb{V}(\cdot)$, number of initial policies $M$, number of policies per iteration $K$, policy parameter space $\Phi$ ;

2 **RESULT:** action to perform $a_t$;

3 $\mathcal{A}_{\text{ME}} = \varnothing$      ▷ Initialize empty collection of policies

4 $\Gamma \leftarrow \text{SAMPLE}(\Phi, M)$      ▷ Sample initial $M$ policies

5 $R, C, \mathcal{T}_{model} = \text{ROLLOUT}(p, s_t, \phi_i, h), \quad \forall \phi_i \in \Gamma$      ▷ Evaluate policies in the model

6 $b_i = f(\phi_i, \mathcal{T}^i_{model}), \quad \forall \phi_i \in \Gamma$ and $\forall \mathcal{T}^i_{model} \in \mathcal{T}_{model}$   ▷ Calculate policies behavior descriptors

7 $\mathcal{A}_{\text{ME}} \leftarrow \phi_i, \quad \forall \phi_i \in \Gamma$      ▷ Store policies in collection

8 **while** $N$ *not depleted* **do**

9    $\Gamma \leftsquigarrow \text{SELECT}(\mathcal{A}_{\text{ME}}, K, \Phi)$      ▷ Select $K$ policies with $C = 0$ from collection

10    $\tilde{\Gamma} \leftsquigarrow \mathbb{V}(\Gamma)$      ▷ Generate new policies

11    $R, C, \mathcal{T}_{model} = \text{ROLLOUT}(p, s_t, \tilde{\phi}_i, h), \quad \forall \tilde{\phi}_i \in \tilde{\Gamma}$    ▷ Evaluate new policies in the model

12    $b_i = f(\tilde{\phi}_i, \mathcal{T}^i_{model}), \quad \forall \tilde{\phi}_i \in \tilde{\Gamma}$ and $\forall \mathcal{T}^i_{model} \in \mathcal{T}_{model}$    ▷ Calculate behavior descriptors

13    $\mathcal{A}_{\text{ME}} \leftarrow \tilde{\phi}_i, \quad \forall \tilde{\phi}_i \in \tilde{\Gamma}$      ▷ Store policies in collection

14 $\Gamma_{\text{lc}} \leftsquigarrow \text{AS}$      ▷ Get policies with lowest cost

15 $\phi_{\text{best}} \leftsquigarrow \Gamma_{\text{lc}}$      ▷ Get policy with highest reward

16 $a_t \leftsquigarrow \phi_{\text{best}}(s_t)$      ▷ Get next action

---

### B.3 PARETO SAFE MAP-ELITES

The Pareto Safe MAP-Elites (PS-ME) planner works similarly to the S-ME one with the exception of the SELECT function at line 9 of Alg. 6. So for this planner we just report the pseudocode of this function in Alg. 7.

---

**Algorithm 7:** SELECT function of PS-ME planner

1 **INPUT:** Collection of policies $\mathcal{A}_{\text{ME}}$, Number of policies to select $K$, Policy parameter space $\Phi$ ;

2 **RESULT:** set of selected policies $\Gamma$;

3 $\Gamma = \varnothing$      ▷ Initialize empty set of selected policies

4 $[\Gamma^0_{\text{ND}}, \dots, \Gamma^n_{\text{ND}}] \leftsquigarrow \text{NON\_DOMINATED\_SORT}(\mathcal{A}_{\text{ME}})$    ▷ Sort collection into non dominated fronts

5 $i = 0$

6 **while** $size(\Gamma) < K$ **do**

7    $\Gamma \leftsquigarrow \Gamma^i_{\text{ND}}$      ▷ Select policies from best front

8    $i = i + 1$

---

## C  ENVIRONMENTS

### C.1  SAFE PENDULUM

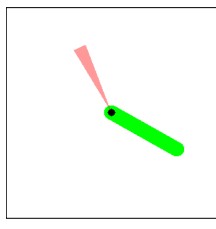

Figure 5: Pendulum upright task.

A safe version of OpenAI's swing up pendulum Luis, in which the unsafe region corresponds to the angles in the $[20°, 30°]$ range, shown in red in Fig. 5. The task consists in swinging the pendulum up without crossing the unsafe region. The agent controls the torque applied to the central joint and receives a reward given by $r = -\theta^2 + 0.1\dot{\theta}^2 + 0.001a^2$, where $\theta$ is the angle of the pendulum and $a$ the action generated by the agent. Every time-step in which $\theta \in [20°, 30°]$ leads to a cost penalty of 1. The state observations consists of the tuple $s = (\cos(\theta), \sin(\theta), \dot{\theta})$. Each episode has a length of $T = 200$.

### C.2  SAFE ACROBOT

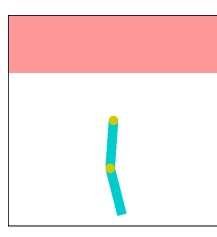

Figure 6: Acrobot upright task.

A safe version of the Acrobot environment (Brockman et al., 2016), shown in Fig. 6. It consists in an underactuated double pendulum in which the agent can control the second joint through discrete torque actions $a = \{-1, 0, 1\}$. The state of the system is observed through six observables $s = (\cos(\theta_0), \sin(\theta_0), \cos(\theta_1), \sin(\theta_1), \dot{\theta}_0, \dot{\theta}_1)$. The reward $r \in [0, 4]$ corresponds to the height of the tip of the double pendulum with respect to the hanging position. The unsafe area corresponds to each point for which the height of the tip of the double pendulum is above 3 with respect to the hanging position, shown in red in Fig. 6. Each episode has a length of $T = 200$ and each time-step spent in the unsafe region leads to a cost penalty of 1.

For this environment, the constraint directly goes against the maximization of the reward. This is similar to many real-world setups in which one performance metric needs to be optimized while being careful not to go out of safety limits. An example of this is an agent controlling the cooling system of a room whose goal is to reduce the total amount of power used while also keeping the temperature under a certain level. The lowest power is used when the temperature is highest, but this would render the room unusable. This means that an equilibrium has to be found between the amount of used power and the temperature of the room, similarly on how the tip of the acrobot has to be as high as possible while still being lower than 3.

### C.3  SAFECAR-GOAL

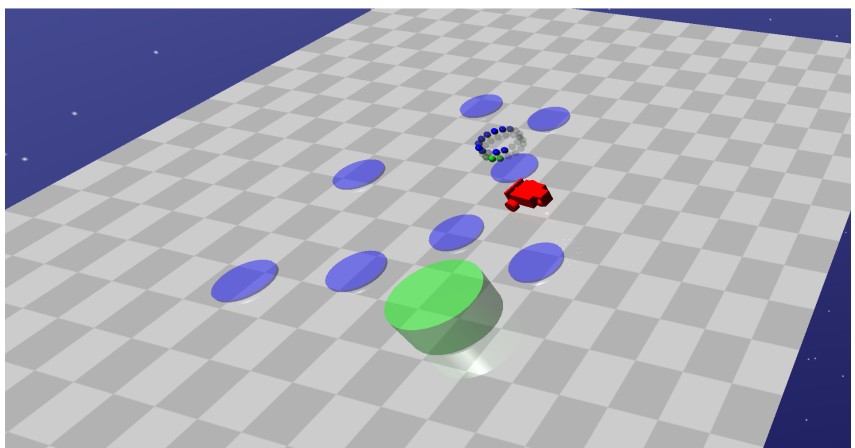

Figure 7: Safety_gym environment

Introduced in Safety Gym (Ray et al., 2019b) it consists of a two-wheeled robot with differential drive that has to reach a goal area in a position randomly selected on the plane, represented as the green cylinder in Fig. 7.(c). When a goal is reached, another one is spawned in a random location. This

repeats until the end of the episode is reached (at 1000 time-steps). On the plane there are multiple unsafe areas, shown as blue circles in Fig. 7.(c), that the robot has to avoid. The placement of these areas is randomly chosen at the beginning of each episode.

The agent can control the robot by setting the wheels speed, with $a \in \mathcal{A} = [-1, 1]$. The observations consists of the data collected from multiple sensors: accelerometer, gyro, velocimeter, magnetometer, a 10 dimensional lidar providing the position of the unsafe areas and the current position of the robot with respect to the goal. This leads to an observation space of size 22.

In the original environment, the cost is computed using the robot position with respect to the position of the different hazards areas. However, as these information are not available in the observations we change the safety function using the Lidar readings and a predefined threshold of 0.9 above which a cost of 1 is incurred.

# D  METRICS

In this appendix we discuss the details on how the metrics used to compare the algorithms are defined.

**Mean Asymptotic Reward (MAR).** Given a trace $\mathcal{T}r$ and a reward $r_t = r(s_t, a_t)$ obtained at each step $t$, we define the mean reward as $R(\mathcal{T}r) = \frac{1}{T}\sum_{t=1}^{T} r_t$. The mean reward in iteration $\tau$ is then $MR(\tau) = R(\mathcal{T}r_t^{(\tau)})$. The measure of asymptotic performance (MAR), is the mean reward in the second half of the epochs (we set $N$ so that the algorithms converge after less than $N/2$ epochs) $MAR = \frac{2}{N}\sum_{\tau=N/2}^{N} MR(\tau)$.

**Mean Reward Convergence Pace (MRCP).** To assess the speed of convergence, we define the MRCP as the number of steps needed to achieve an environment-specific reward threshold $r_{thr}$. The unit of MRCP($r_{thr}$) is real-system access steps, to make it invariant to epoch length, and because it better translates the sample efficiency of the different methods.

**Probability percentage of unsafe** ($p(\textbf{unsafe})[\%]$). To compare the safety cost of the different algorithms, we compute the probability percentage of being unsafe during each episode as $p(\text{unsafe}) = 100 * \frac{1}{T}\sum_{k=0}^{T} \mathcal{C}_k$ where $T$ is the number of steps per episode. We also compute the *transient* probability percentage $p(\text{unsafe})_{trans}[\%]$ as a measure to evaluate safety at the beginning of the training phase, usually the riskiest part of the training process. It is computed by taking the mean of $p(\text{unsafe})$ on the first 15% training epochs.

# E  TRAINING AND HYPER-PARAMETERS SELECTION

All the training of model-based and model-free methods have been perform in parallel with 6 CPU servers. Each server had 16 Intel(R) Xeon(R) Gold CPU's and 32 gigabytes of RAM.

## E.1  MODEL BASED METHOD

For the Safe Acrobot and Safe Pendulumn environments we used as model $p$ a deterministic deep auto-regressive mixture density network ($DARMDN_{det}$) while for the Safety-gym environment we used the same architecture but without auto-regressivity ($DMDN_{det}$).

Table 2: Model and Agent Hyper-parameters

|  |  | **Safe Pendulum** | **Safe Acrobot** | **SafeCar-Goal** |
|---|---|---|---|---|
|  | Model | | | |
| | Optimizer | Adam | Adam | Adam |
| | Learning rate | 1e-3 | 1e-3 | 1e-3 |
| $D(AR)MDN_{det}$ | Nb layers | 2 | 2 | 2 |
| | Neurons per layer | 50 | 50 | 50 |
| | Nb epochs | 300 | 300 | 300 |
|  | Planning Agents | | | |
| CEM and RCEM | Horizon | 10 | 10 | 30 |
| | Nb actions sequence | 20 | 20 | 3000 |
| | Nb elites | 10 | 10 | 12 |
| S-RS and RS | Horizon | 10 | 10 | 30 |
| | Nb actions sequence | 100 | 100 | 3000 |
| ME, S-ME, PS-ME | Horizon | 10 | 10 | 30 |
| | Nb policies | 100 | 100 | 525 |
| | Nb initial policies | 25 | 25 | 25 |
| | Nb policies per iteration | 5 | 5 | 10 |
| | Behavior space grid size | $50 \times 50$ | $50 \times 50$ | $20 \times 20$ |
| | Nb policy params. | 26 | 83 | 77 |
| | Nb policy hidden layers | 1 | 2 | 1 |
| | Nb policy hidden size | 5 | 5 | 3 |
| | Policy activation func. | Sigmoid | Sigmoid | Sigmoid |

## E.2  MODEL FREE METHOD

All model free algorithms implementation and hyper-parameters were taken from the https://github.com/openai/safety-starter-agents repository.

## F OPTIMALITY

Safe-RL algorithms have to optimize the reward while minimizing the cost. Choosing which algorithm is the best is a multi-objective optimization problem. Fig. 8 shows where each of the methods tested in this paper resides with respect to the MAR score and the p(unsafe). Methods labeled with the same number belong to the same optimality front with respect to the two metrics. A lower label number indicates an higher performance of the method.

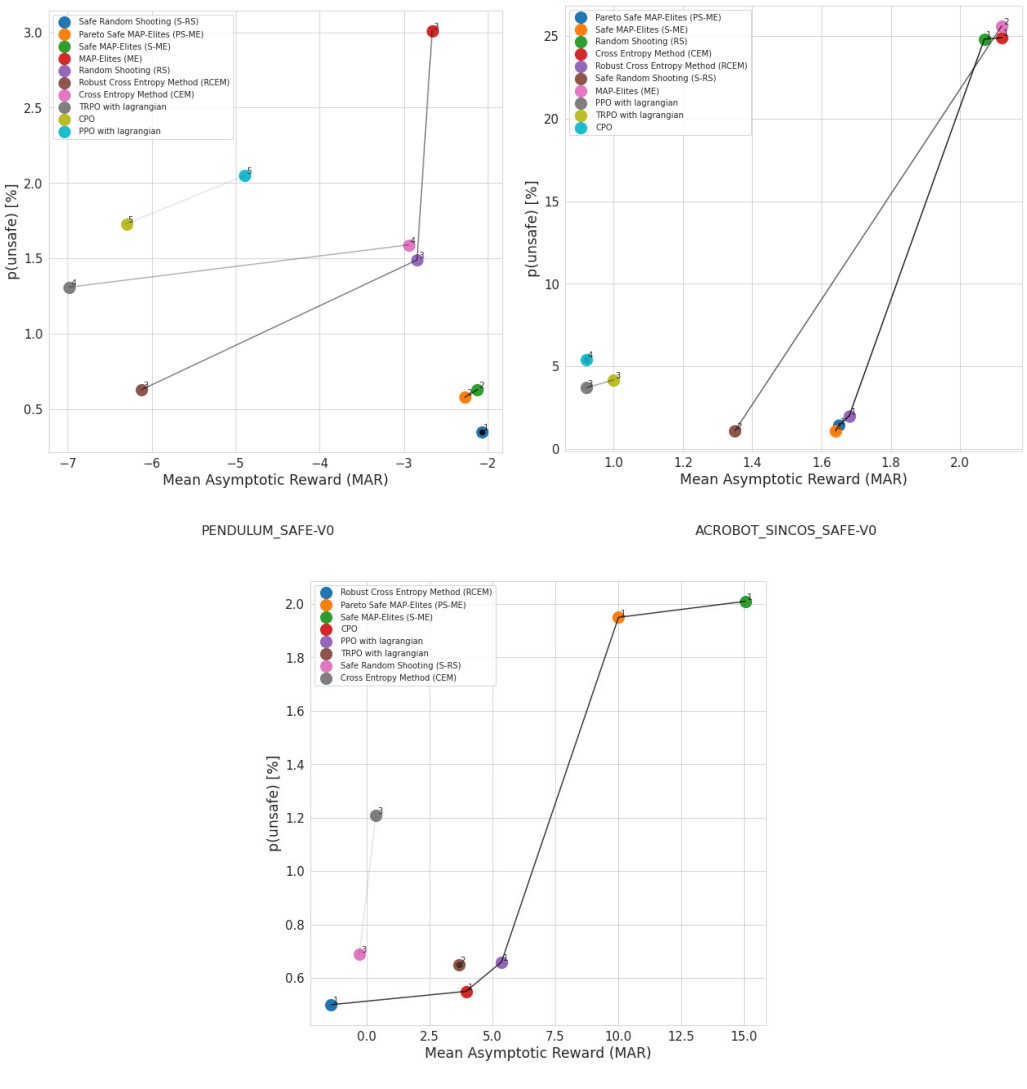

Figure 8

## G EXPLORATION PLOTS

In this section we present plots about the explored state distribution for the different algorithms on the two environments.

The *State Coverage* plots, leftmost one in the figures, represents the density of times a state has been explored. The unsafe area is highlighted in red.

The *Reward Heatmap* plots, center ones, show the distribution of the visited states overimposed to the reward landscape. The unsafe area is highlighted in red.

The *Reward bins* plots, rightmost ones, show the histogram of visited stated with respect to the reward of that state. The histograms are generated by dividing the interval between the minimum and maximum reward possible in the environment into 10 buckets and counting how many visited states obtain that reward.

It is possible to see on the reward plots for the Safe Acrobot environment in Appendix G.2 how the safety constraint limits the number of times the states with $r > 3$ are visited by safe methods compared to unsafe ones.

## G.1    SAFE PENDULUM

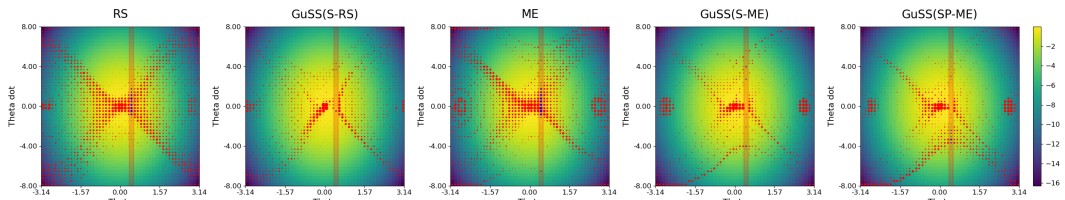

Figure 9: Explored states for Safe Pendulum. For Safe Pendulum a state $s$ corresponds to the angular position and velocity of the pendulum $(\theta, \dot{\theta})$. The states are color coded according to the corresponding reward. The red-shaded area corresponds to the unsafe area. Each point represent an explored state. Red points are safe states while blue points are unsafe ones. The size of the point is proportional to the number of times that state has been visited.

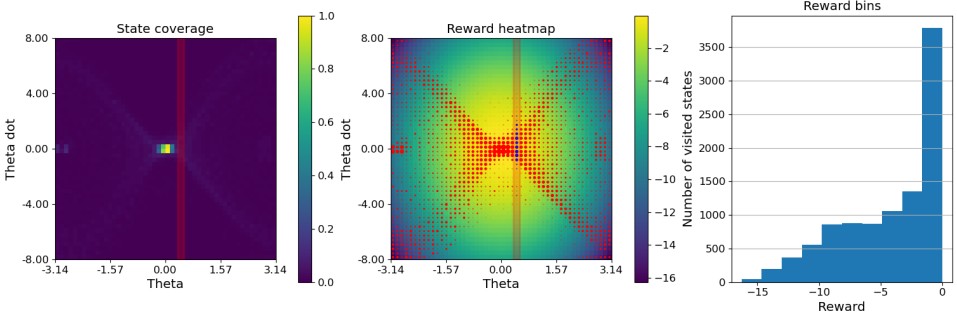

Figure 10: Exploration plot for RS on Safe Pendulum.

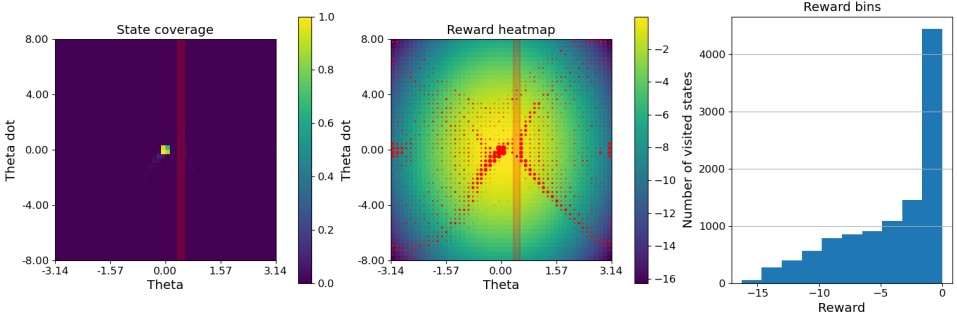

Figure 11: Exploration plot for GuSS(S-RS) on Safe Pendulum.

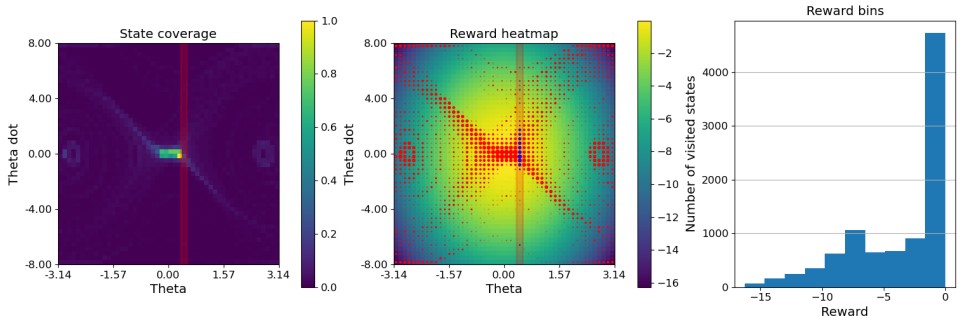

Figure 12: Exploration plot for ME on Safe Pendulum.

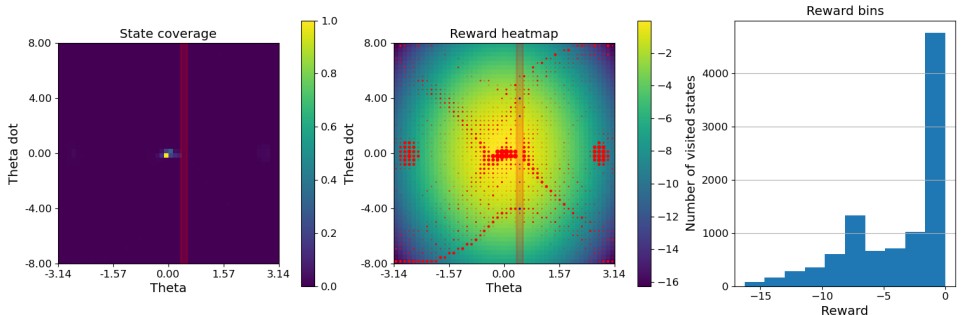

Figure 13: Exploration plot for GuSS(S-ME) on Safe Pendulum.

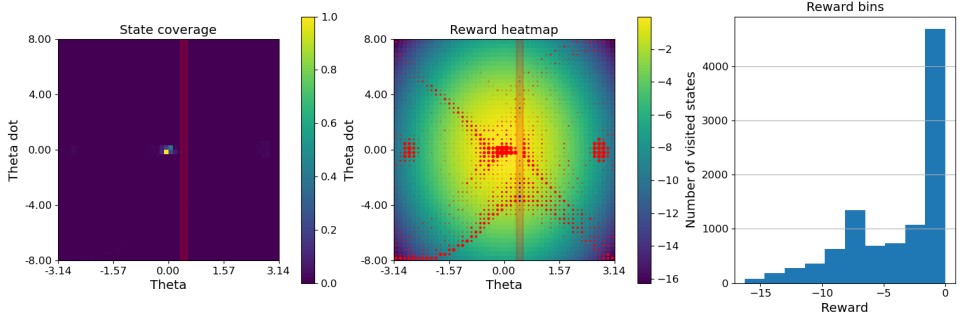

Figure 14: Exploration plot for GuSS(PS-ME) on Safe Pendulum.

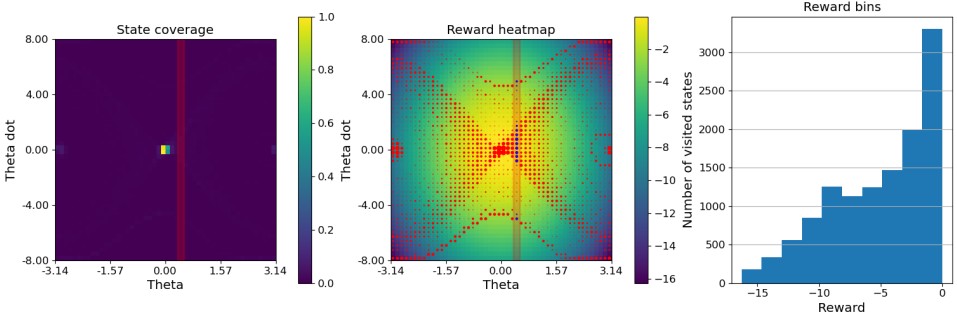

Figure 15: Exploration plot for CEM on Safe Pendulum.

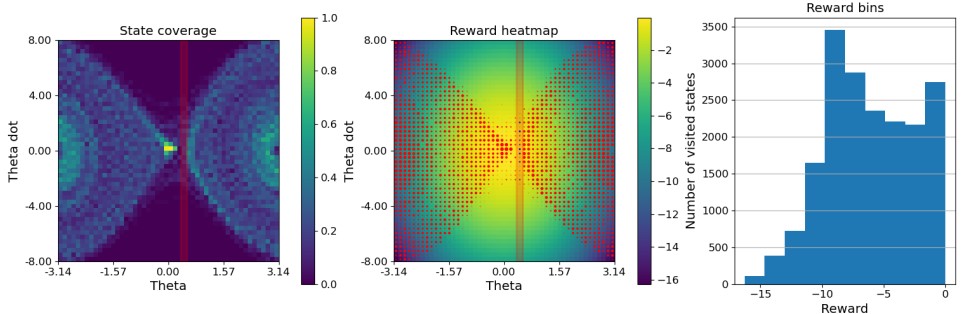

Figure 16: Exploration plot for RCEM on Safe Pendulum.

## G.2 SAFE ACROBOT

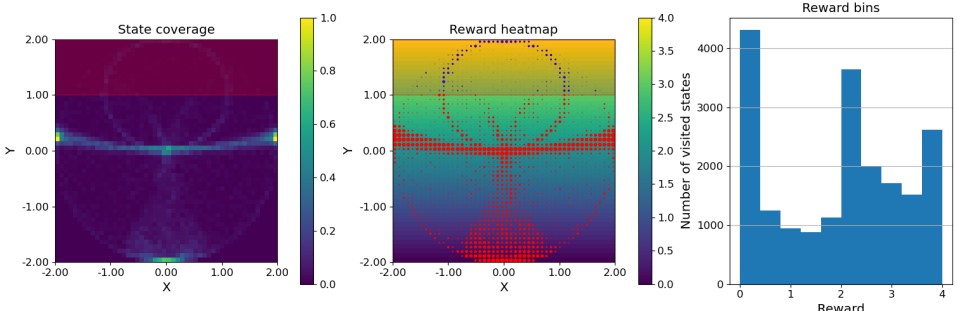

Figure 17: Exploration plot for RS on Safe Acrobot.

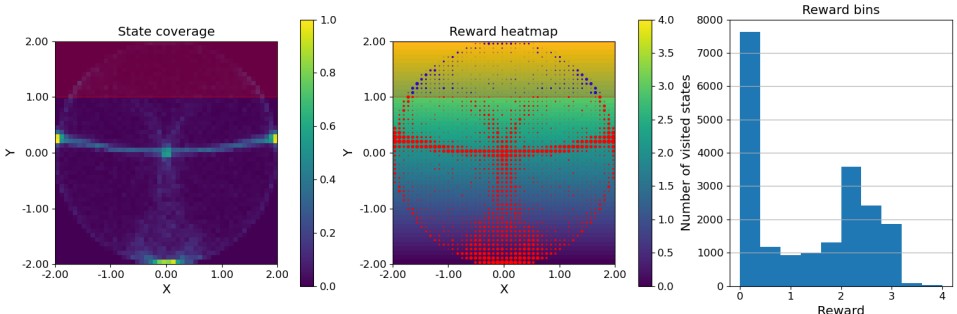

Figure 18: Exploration plot for GuSS(S-RS) on Safe Acrobot.

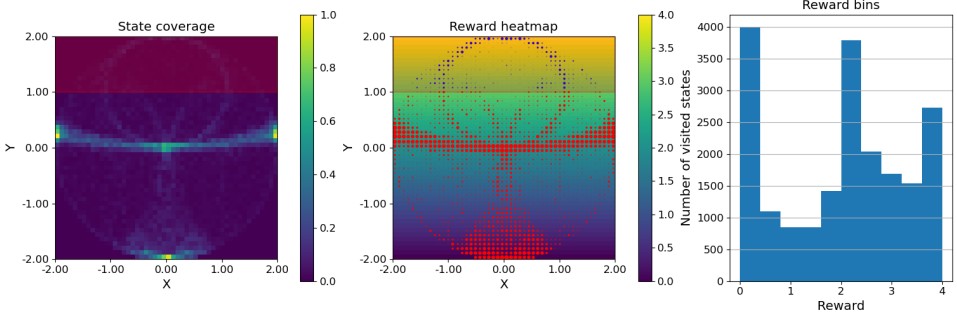

Figure 19: Exploration plot for ME on Safe Acrobot.

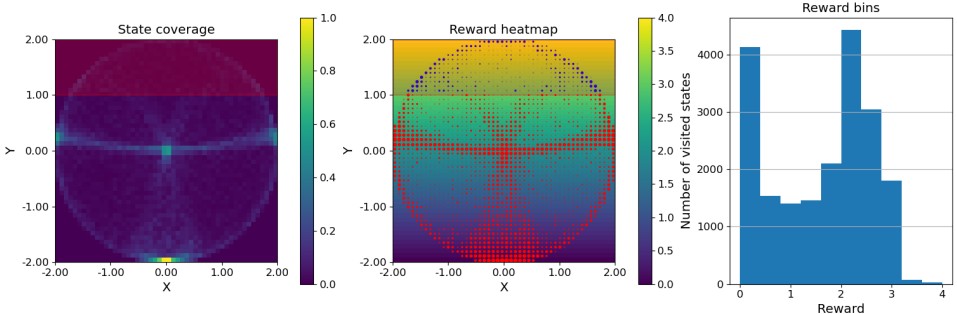

Figure 20: Exploration plot for GuSS(S-ME) on Safe Acrobot.

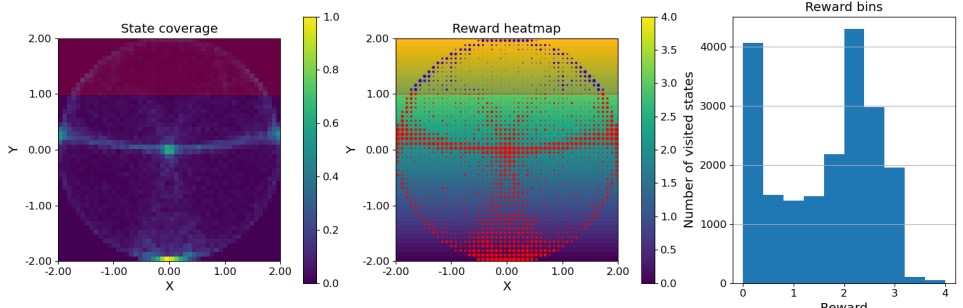

Figure 21: Exploration plot for GuSS(PS-ME) on Safe Acrobot.

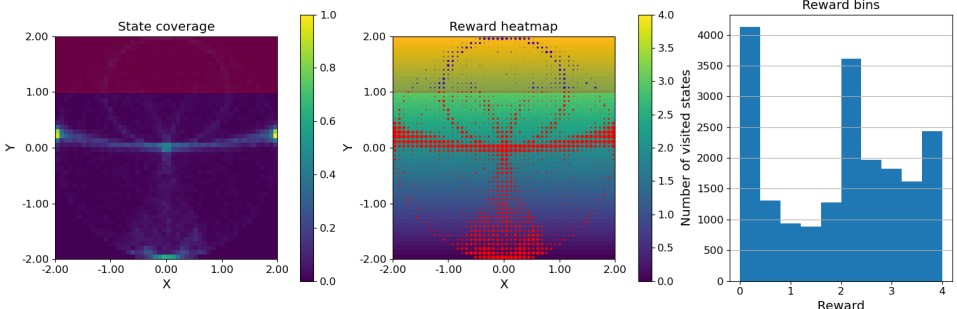

Figure 22: Exploration plot for CEM on Safe Acrobot.

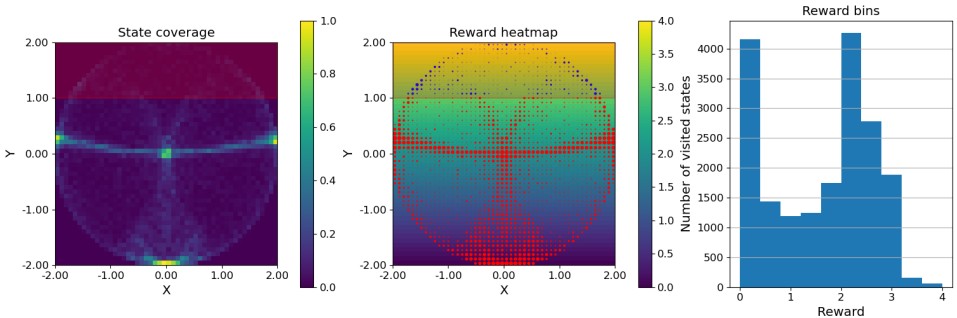

Figure 23: Exploration plot for RCEM on Safe Acrobot.

