# OpenReview forum: "Guided Safe Shooting: model based reinforcement learning with safety constraints"
_ICLR.cc/2023/Conference — Submitted to ICLR 2023_

### Official Review · Reviewer_G6tK · 2022-10-24

**Confidence:** 3
**Correctness:** 2
**Technical Novelty And Significance:** 3
**Empirical Novelty And Significance:** 2
**Recommendation:** 3

**Clarity, Quality, Novelty And Reproducibility:**


The presentation of the method is very short, and many descriptions are only in the appendix, while the paper spends a long time on presenting some of the results. This makes it complicated to follow the presentation and grasp the working of the methods, as well as understand all the results presented, since the ones in the appendix are without or only with a short description.
I don't think it would be possible to reimplement the method just from the description of the paper.

It is unclear to me what exactly the behavior space in MAP elites is, what characterizes a policy behavior in this paper?

Of the multiple proposed methods, what is the recommendation of which method to use in which scenario?

**Strength And Weaknesses:**

The use of quality diversity methods is an interesting approach to increase the exploration of the agent, which is an important aspect to collect information about the safety conditions in every part of the state space and to collect alternative planning configurations depending on the policy behaviour.
The experiments show some of the variants to be competitive with the baseline methods, although not consistently, they however still outperform the model-free alternatives, as we should expect.


**Summary Of The Paper:**

The paper proposes a new method for safe model-based RL that minimizes safety constraint violations during planning.
Three variants of the method are proposed (S-RS, S-ME, PS-ME) and experiments are performed to show their effectiveness in terms of safety, state space exploration, and rewards earned.

**Summary Of The Review:**

The paper works in an interesting direction and shows some promising results, but the method itself and how it works is unclear. Several methods are proposed, but without a clear winner or recommendation of which method to apply for future work.

As it is, I do not think the paper is sufficiently ready for publication.

---

### Official Review · Reviewer_WaLB · 2022-10-24

**Confidence:** 4
**Correctness:** 4
**Technical Novelty And Significance:** 2
**Empirical Novelty And Significance:** 2
**Recommendation:** 3

**Clarity, Quality, Novelty And Reproducibility:**

Overall the paper suffers from lack of clarity, as to what is the real contribution. I think that methodologically there's no significant contribution of the paper - but the novelty factor perhaps comes from the experiments and how they're set up (e.g. randomized safety constraints). If that is indeed the case, maybe the authors should refocus the paper - title, abstract and conclusions need to be adapted accordingly.

Some minor comments below:

* Using discounted return is problematic for safety-critical applications on real systems without a clear horizon.
* "The model is trained and evaluated on the real-system" is misleading as the experiments are simulation only.
* page 4: use -> learn?
* Initial rollouts in Algorithm 1 are unsafe.
* What distinguishes "deterministic deep mixture density networks" as a good choice for safety-critical control applications?
* How do you fix N and h in your planners? Is it critical?
* What is the motivation of introducing a Pareto-Safe version of S-ME in Section 4.2.3? Please discuss.
* TRPO lag and PPO lag stand for I guess TRPO/PPO used with Lagrangian relaxation?
* What is meant by "behaviour space" is not clear to me, do you mean the specific way the QD-approaches evaluate and pick the rolled-out trajectories?

**Strength And Weaknesses:**

Strengths:

* The introduced QD-planners are extensively tested in 3 different environments against several competing approaches.

Weaknesses:

* The paper does not introduce a new method per se, as in a new model-based RL algorithm that e.g. learns a model in a particular way, or that implements a new model structure. Rather, three ad-hoc QD-planners are introduced but not analyzed rigorously/theoretically: the heuristics are shown to work well in the chosen experimental scenarios, but what are the failure points? When would they be suboptimal and when could the resulting policies violate safety constraints? These are not discussed theoretically or verbally.

* As the authors themselves mention at the conclusion, "if the model is wrong, it could easily lead the agent to unsafe states". Indeed the focus of model-based RL approaches introduced to safety critical environments, need to consider very critically how the learned model could mislead the agent, or how to learn robust models that do not do so. I would suggest the authors to look up the literature on robust control or the more recent uncertainty-aware learning controllers (using GPs, etc.).

* If the contribution of the paper is not a new model-based RL agent but rather the custom planners, then as mentioned above, they would need to be analyzed critically and rigorously, e.g. as is often done in the MPC literature. I would suggest the authors to look up e.g. stochastic/robust/learning MPC formulation.

**Summary Of The Paper:**

The paper presents a model based safe-RL algorithm that uses custom planners belonging to the family of Quality Diversity methods. Three such custom planners are proposed. The resulting RL approach is tested in three different simulation setups, with randomized safety constraints.

**Summary Of The Review:**

Overall the paper should be rejected as it is lacking a clear methodological contribution, see comments above.

---

### Official Review · Reviewer_5gos · 2022-10-24

**Confidence:** 4
**Correctness:** 3
**Technical Novelty And Significance:** 2
**Empirical Novelty And Significance:** 3
**Recommendation:** 5

**Clarity, Quality, Novelty And Reproducibility:**

The fonts in the graphics are too small. Also the line colors are not easy to tell apart.

What means the bolding in Table 1? It seems that some of the numbers would not be significant. You use 3 and 5 seeds and then compute 90% confidence intervals, which is likely not very trustworthy.

Code is provided and details are given, so I think reproducibility is high.

Novelty: The paper is novel for the Map Elite shooting method being applied to MBRL

**Strength And Weaknesses:**

Strengths:
- important subject
- bringing quality-diversity planners into MBRL

Weaknesses:
- missing baselines
- no statistical analysis
- only toy environments
- not really clear which improvement the decisive one
- too small fonts in the graphics

Details:
Since you are not dealing with uncertainty while planning, I am wondering what about a standard approach where the costs term is simply substracted from the reward with a big constant factor (referred to as "big M" approach in control). This should be a baseline: so ME with this r - M*c as a reward with large M. Then we would see whether the particular optimization improvements that you do really are crucial. How many rounds of ME are you performing, every timestep. Is there any shift initialiation etc?

I suspect that most of the performance gain of the paper comes from the diversity term in the ME optimizer. However, there was prior work [1] that does all of this already: epistemic uncertainty to collect data needed for model learning, safety awareness including aleatoric uncertainty (which is not so important here, as you have only deterministic and simple environments) and probabistic safety constraints, also in highdimensional control problems.

I cannot find the description of the "behavior space" used.

[1] Risk-Averse Zero-Order Trajectory Optimization, CoRL 2021
https://openreview.net/forum?id=WqUl7sNkDre

**Summary Of The Paper:**

The paper proposes a method to obtain safe model-based RL by using a guided safe shooting method (with variations) and applies it to 3 toy tasks.

The paper a quality-diversity method as an optimization method method with some variations for safe planning.

**Summary Of The Review:**

The paper looks at an important problem, but in the current state, as the paper is empirical, the evaluation is limited to pretty simple environments. Also a simple baseline, for identifying which part is really important is missing. Apart from that, there is a prior work that was not mentioned that solves the tackled problem.

---

### Official Review · Reviewer_uMvM · 2022-10-27

**Confidence:** 3
**Correctness:** 3
**Technical Novelty And Significance:** 2
**Empirical Novelty And Significance:** 3
**Recommendation:** 3

**Clarity, Quality, Novelty And Reproducibility:**

I found the paper largely easy to read, and the approach is clearly defined. Taking a model-based approach to safe RL has been discussed at lengths in the literature, and is well-motivated. The results seem solid, and likely they are possible to be reproduced.

I am however strongly concerned about the novelty. A simple google search reveals a number of papers that combine model-predictive control with (safe) reinforcement learning, and none of these papers are cited. Moreover, a paper that is cited [Liu et al ]and declared very related, is not discussed in detail. From the current status, I tend to say that the novelty is absolutely unclear, but I will be happy to read the answers of the authors and re-evaluate my assessment. See below a number of papers that I found very quickly, but I am sure that there are more, especially in the controls community.

Samuel Pfrommer, Tanmay Gautam, Alec Zhou, Somayeh Sojoudi:
Safe Reinforcement Learning with Chance-constrained Model Predictive Control. L4DC 2022: 291-303

Torsten Koller, Felix Berkenkamp, Matteo Turchetta, Andreas Krause:
Learning-based Model Predictive Control for Safe Exploration and Reinforcement Learning. CoRR abs/1803.08287 (2018)

Alexandre Didier, Kim Peter Wabersich, Melanie N. Zeilinger:
Adaptive Model Predictive Safety Certification for Learning-based Control. CDC 2021: 809-815

As a further remark to the authors, they should consider discussing the area of ‘shielded RL’, which naturally takes a model-based approach. See (only some of the many) references below.

Mohammed Alshiekh, Roderick Bloem, Rüdiger Ehlers, Bettina Könighofer, Scott Niekum, Ufuk Topcu:
Safe Reinforcement Learning via Shielding. AAAI 2018: 2669-2678

Nils Jansen, Bettina Könighofer, Sebastian Junges, Alex Serban, Roderick Bloem:
Safe Reinforcement Learning Using Probabilistic Shields. CONCUR 2020: 3:1-3:16


**Strength And Weaknesses:**

Strengths

- The topic is very relevant and well-fitting for ICLR.
- The concept of combining model-based control with RL is natural.
- The evaluation considers state-of-the-art environments.

Weaknesses
- The novelty of the approach is unclear, and the related work is not discussed appropriately.
- The paper seems a bit unpolished at parts.



**Summary Of The Paper:**

This paper takes a particular view on safe reinforcement learning in the form of model-predictive control. In particular, the authors present an approach, called ‘guided safe shooting (GuSS), that combines model-based planning with RL, towards a minimal violation of safety. Technically, a constrained MDP approach is realized, together with three different planning methods. The approach is evaluated by means of three OpenAI environments.

**Summary Of The Review:**

Good idea and approach, but the novelty is in doubt.

---

### Author Response · Authors · 2022-11-19
**Response to reviewers**

We thank the reviewers for their insightful suggestions and helpful comments. They will greatly help in improving the paper for the future.

---

### Decision · Program_Chairs · 2023-01-20

**Decision:**

Reject

**Justification For Why Not Higher Score:**

Limited contribution

**Justification For Why Not Lower Score:**

N/A

**Metareview: Summary, Strengths And Weaknesses:**

In this work, the authors identify safe deployment of policies learned by RL as a challenge and propose Guided Safe Shooting as an approach to minimize violations of safety constraints.  The core premise of the approach involves learning a model and avoiding unsafe actions by reference to planning in the learned model.  Different planning approaches are considered.

The reviews leaned towards rejection.  A common theme in the reviews is that the problem framing is interesting and well motivated, but the actual contribution is likely not novel, or at the very least, it is unclear what part of it is novel relative to the existing body of literature on this topic.

Building on these reviewer concerns, it is my sense that the authors have sort of re-proposed the default approach for safe model-based RL.  Perhaps a bit too cursory of a literature search was undertaken before setting out on this work, since there is a pretty large published literature on this problem.  While in this work, the approach is shown to be effective on simple environments, the authors themselves note in the discussion that this approach can fail from a safety standpoint if the models are wrong.  They also suggest that e.g. model uncertainty could be used to be more conservative.  As far as I am aware, these issues have previously been identified, and these and many other modifications have already been explored in the literature.